# Cloud Fraction estimation using Random Forest classifier on Sky Images

Sougat Kumar Sarangi<sup>1</sup>, Chandan Sarangi<sup>1</sup>, Niravkumar Patel<sup>2</sup>, Bomidi Lakshmi Madhavan<sup>3</sup>, Shantikumar Singh Ningombam<sup>4</sup>, Belur Ravindra<sup>4</sup> and M. Venkat Ratnam<sup>3</sup>

- <sup>1</sup> Department of Civil Engineering, Indian Institute of Technology Madras, Chennai, India
  - <sup>2</sup> Department of Engineering Design, Indian Institute of Technology Madras, Chennai, India
  - <sup>3</sup> Aerosols, Radiation and Trace Gases Group, National Atmospheric Research Laboratory, Gadanki, India
  - <sup>4</sup> Sun and Solar System Group, Indian Institute of Astrophysics, Bangalore, India

Correspondence to: Chandan Sarangi (chandansarangi@civil.iitm.ac.in)

10 **Abstract.** Cloud fraction (CF) is an integral aspect of weather and radiation forecasting, but real time monitoring of CF is still inaccurate, expensive and exclusive to commercial sky imagers. Traditional cloud segmentation methods, which often rely on empirically determined threshold values, struggle under complex atmospheric and cloud conditions. This study investigates the use of a Random Forest (RF) classifier for pixel-wise cloud segmentation using a dataset of semantically annotated images from five geographically diverse locations. The RF model was trained on diverse sky conditions and atmospheric loads, ensuring robust performance across varied environments. The accuracy score was always above 85% for all the locations along with similarly high F1 score and Receiver Operating Characteristic - Area Under the Curve (ROC-AUC) score establishing the efficiency of the model. Validation experiments conducted at three Atmospheric Radiation Measurement (ARM) sites and two Indian locations, including Gadanki and Merak, demonstrated that the RF classifier outperformed conventional Total Sky Imager (TSI) methods, particularly in high-pollution areas. The model effectively captured long-term weather and cloud patterns, exhibiting strong location-agnostic performance. However, challenges in distinguishing sun glares and cirrus clouds due to annotation limitations were noted. Despite these minor issues, the RF classifier shows significant promise for accurate and adaptable cloud cover estimation, making it a valuable tool in climate studies.

#### 1 Introduction

Clouds are a fundamental constituent of our weather systems and one of the most critical climate variables influencing the Earth's radiation budget. Cloud albedo influences the amount of solar radiation reflected into space and hence affects the energy budget at Earth's surface and in the atmosphere (Ramanathan et al., 1989). It also influences the atmospheric thermodynamics, surface fluxes and hence the water vapor and carbon cycle (Várnai and Marshak, 2015), thereby impacting the extent of many land-atmosphere processes, feedback and interactions at various spatio-temporal scales. Consequently, the scientific community requires specific devices to observe the fluctuations in cloud cover and other cloud properties at a high

spatial and temporal resolution. Typically, these devices fall into two categories: satellite-based and ground-based imagers. Satellite imagers observe clouds over larger spatial domains (Verma et al., 2018) with temporal resolutions as high as 10minutes (Huang et al., 2019). Ground-based sky monitoring devices, on the other hand, capture data at high temporal resolutions, ranging from as frequent as 30 seconds to 5 minutes over a fixed point (Nouri et al., 2019).

Over the years, researchers have developed numerous algorithms to detect clouds in images and classify them into broader categories of cloud types. These cloud detection algorithms primarily fall into two categories: thresholding techniques and classifier-based methods. The clear sky (CSL) threshold method, as outlined by (Shields et al., 2009), uses spectral information—particularly from the red and blue bands—to differentiate between cloudy and clear-sky conditions. This technique has been widely adopted by researchers (Chauvin et al., 2015; Chow et al., 2011; Ghonima et al., 2012; Kuhn et al., 2018; Lothon et al., 2019). However, a notable limitation is that the threshold value can vary across an image, influenced by the relative distance between the sun and each pixel in the image. This dynamic adjustment is crucial because scattering properties change with variations in the path length and the angular position of the sun, as demonstrated by (Long et al., 2006). As such, an adaptive thresholding technique was proposed based on distance from the sun (Li et al., 2011; Yang et al., 2012). However, cloud images are inherently diverse, featuring complex spectral information. Due to this diversity, conventional image segmentation techniques, such as thresholding and shape differentiation methods, struggle to provide precise and consistent segmentation results.

Modern algorithms integrate multiple features into building a classifier, including spectral, statistical, and Fourier-transformed features, in a supervised manner (Calbó et al., 2008). Many supervised algorithms have been used for recognising different cloud types. (Heinle et al., 2010) and (Rajini and Tamilpavai, 2018) have used a k-nearest neighbour (KNN) classifier to determine cloud type using statistical features. (Kazantzidis et al., 2012) proposed an improved KNN classifier for cloud-type determination where solar zenith angle and visible solar disk were considered. To improve the speed of classification, (Rajini and Tamilpavai, 2018) have used neighbourhood component analysis to optimize the feature selection. (Li et al., 2015) have established a cloud identification model based on the Otsu technique (Otsu, 1979) with the aim of increasing the accuracy of short-term solar power production. (Satilmis et al., 2020) have developed a hierarchical histogram merging method to classify cloud types in high dynamic range (HDR) images. While most authors have predominantly used RGB colour space or some derivative feature of RGB values, (Jayadevan et al., 2015) have suggested the use of hue-saturation-value colour space to increase the contrast between clouds and background sky. The use of machine learning for cloud classification has gained significant traction in recent years. (Taravat et al., 2015) and (Li et al., 2016) showcase the use of multi-layer perceptron neural networks as well as support vector machines in cloud detection. Artificial neural networks (ANN) have also been implemented to distinguish clouds from clear sky (Xia et al., 2015) using a hybrid KNN and ANN method. (Kliangsuwan and Heednacram, 2018) introduced using Fourier-transformed features for classification using ANN. (Wan et al., 2020) combined several texture, colour and spectrum features to classify clouds as cirrus, cumulus, and stratus clouds. While many existing methods excel at classifying different cloud types, their accuracy tends to hover around 80-85% when it comes to precisely identifying

individual cloud pixels. A U-Net based convolutional neural network model developed by (Fa et al., 2019), (Fabel et al., 2022) and (Xie et al., 2020) have shown promising results of having nearly the same cloud fraction output as obtained by manual observations. An improvement in the encoder-decoder model has been developed by (Ye et al., 2022) where the decoding stage is branched into binary segmentation for cloud detection and attribute discrimination and feature learning. However, these CNN techniques require high-power graphical processing units to process.

This paper's primary focus is on presenting a high accuracy cloud fraction retrieval approach that leverages the power of random forest for pixel-level cloud classification in sky images. The predicted cloud fractions are compared with semantically annotated sky images from five different locations with varying atmospheric and sky conditions to validate their accuracy and reliability. Moreover, a baseline comparison has been done between the output of the total sky imager used at the three ARM sites and our model's output. A yearly comparison of trends in observed cloud fraction has also been conducted to demonstrate the stability of the classifier's output under different climatic conditions. To support these objectives, the remainder of the paper is structured as follows. Section 2 describes the data used in this study, including observing sites, datasets, preprocessing steps, and the creation of ground truth masks. Section 3 details the model selection process. Section 4 covers model training and evaluation, with Section 4.1 focusing on model validation and Section 4.2 discussing the application of the Random Forest classifier over the Merak site. Finally, Section 5 presents the conclusions drawn from this study.

#### 2 Data

70

## 2.1. Observing sites and datasets

The sky image data used in the current work are taken from three different ARM sites (Flynn & Morris, tsi sky imager (a1)). These are publicly available data with the following sites: the Black Forest, Germany (FKB; 48.54° N, 8.40° E, 511 masl); Southern Great Plains, Central Facility, Lamont, Canada (SGP; 36.61° N, 97.49° W, 315 masl); and Tropical Western Pacific, Central Facility, Darwin, Australia (TWP; 12.42° S, 130.89° E, 30 masl). We also took sky image data from National Atmospheric Research Laboratory (NARL; 13.48° N, 79.18° E, 375 masl) Gadanki, India. All four sites utilize the same type of instrument - a Total Sky Imager (TSI) - for capturing sky images. The TSI, used at all four locations, features a domeshaped, spherical mirror with a 180° field-of-view of the sky. A downward-facing CCD camera is placed above the dome mirror to take images. To prevent image saturation from direct sunlight, a rotating shadow band is used to track and block the sun continuously. This multi-site data collection enables the evaluation of the model's robustness across various locations and their corresponding atmospheric conditions.

Additionally, multi-year sky image data is taken from an all-sky camera (Ms. Prede, Japan) recorded for every 5 minutes interval from the National Large Solar Telescope site, Merak (33.80° N; 78.62° E; 4310 m, asl), Ladakh, India. Such cloud data in the high-altitude mountain sites in the Ladakh region is used for the astronomical site characterization program of the Indian Institute of Astrophysics, Bengaluru, India. There are several unique features of the observing site, such as low aerosol

content (~0.05 at 500 nm, (Ningombam et al., 2015)), with 61-68% of clear skies in a year (Ningombam et al., 2021), dry and cold atmospheric conditions, located in the rain shadow area of the Himalaya. A comprehensive statistical analysis is conducted on the model's output for this site to show the effectiveness of capturing some of the intricate atmospheric conditions of this location.

#### 2.2. Preprocessing

All sky images are organized using a timestamp-based naming convention to maintain chronological order. Any image captured before 6:00 AM and after 6:00 PM (local time) were removed because of bad lighting conditions. Additionally, images captured during rain were manually removed because of the undue distortions caused by the raindrops on the lens. Since images from different locations had varying image size, all the images were cropped and resized to 280 x 280 pixels to remove dead zones in the image. The images had lens glares and occasional occlusions from nearby structures and instruments. To mitigate these issues, a circular mask of radius 130 pixels is applied to all images, effectively removing potential interferences that could disrupt the training process. An example of one such pre-processed image is shown in Fig. S1 of the Supplementary document.

#### 2.3. Selection

A large part of the uncertainty in this kind of supervised training is determining the optimal level of variability within the dataset to ensure the model learns effectively. A well-curated selection must be made, encompassing various scenarios, including clear skies, different cloud cover percentages and atmospheric/sky conditions for the ML model to understand the diverse data and increase its robustness. A systematic approach was followed to curate the image dataset for training our machine learning model. Initially, a thorough visual inspection of the entire image pool was performed. The goal was to ensure a balanced representation of cloud cover percentages in our dataset. Around 300 images from each site were carefully selected, ranging from no clouds to 100% cloud coverage (based on visual estimation). This approach allowed us to create a diverse and well-structured training dataset of different CF and different cloud types required for training. This would be essential in developing an effective machine-learning model for cloud cover classification. A set of 100 images, which also contains various cloud percentages between 0 to 100% and cloud types, was kept aside for validation of the model's training.

### 2.4. Ground truth masks

About 2000 sky images, selected from all five locations, are meticulously annotated using the MATLAB image labeller app (The MathWorks Inc., 2022). This tool offers advanced capabilities for image annotation, allowing for annotations in the form of lines, rectangles, polygons, or pixel-level detail, with the added benefit of colour coding for a well-organized graphic user interface.

For images with complex cloud shapes, pixel-level annotation is the optimal selection. These annotations involve assigning numerical labels to different elements in the images, where 0 represents the sky, 1 represents the sun, 2 represents the clouds, and 3 represents occlusions. Given the diverse and complex nature of clouds, along with variations in experts' perspectives on cloud pixels within an image, the annotation process involves three different domain experts. An overlap of their annotations is taken to produce the final annotated image.

These annotated images are saved as separate pixel matrix files, retaining the same name as the original image file. This is a crucial step to ensure that the correct annotations are cross-referenced during the model's training and testing phases.

#### 3 Model Selection

RF is a machine-learning technique to solve classification problems (Breiman, 2001). It is an ensemble method that combines the predictions of multiple decision trees to produce a more accurate and stable prediction. Here at every instance, a node is partitioned based on one optimal feature among several selected features. Hence, for each decision tree, there is maximum independence leading to generalized performance and a decreased chance of over-fitting (Dietterich, 2000). The final prediction is a result of the majority vote calculated using the probability of each kth class:

$$P_k = \frac{w_k N_k}{\sum_{j=1}^m w_j N_j} \tag{1}$$

Where m is the total number of classes,  $N_j$ ,  $N_k$  are the number of trees predicting the  $j^{th}$  and  $k^{th}$  class and  $w_j$ ,  $w_k$  are the weights of the  $j^{th}$  and  $k^{th}$  class. For tuning the RF classifier, two of the most important parameters that need to be effectively chosen are number of decision trees ( $N_{tree}$ ) and number of selected feature variables ( $M_{feat}$ ). Higher  $M_{feat}$  implies an increased correlation between two decision trees resulting in poor categorization. Similarly, larger  $N_{tree}$  can provide increased accuracy at the cost of higher computational resources. It has been found that higher  $N_{tree}$  can lead to over-fitting in some cases (Scornet Erwan, 2017). (Fu et al., 2019) and (Ghasemian and Akhoondzadeh, 2018) have suggested choosing the two parameters such that they are large enough to capture the patterns and have a wide diversification, but small enough for the model to run at reduced computational power and prevent overfitting.

A key limitation of Random Forests is that, due to their ensemble nature, it is difficult to trace individual pixel-level classifications back to specific features or decision paths. Even then, (Mu et al., 2017) explain that RF has lower time and computation costs when the data size is larger than most machine learning algorithms. (Wang et al., 2020) have used RF for cloud masking and study of cloud thermodynamics using satellite data which have shown good resemblance with the lidar observations. (Sedlar et al., 2021) have classified cloud types based on surface radiation measurements using RF. (Li et al., 2022) have used RF to classify cloud types from images taken by an all-sky imager for astronomical observatory site selections.

While these findings demonstrate the prowess of RF in cloud type classification, they also serve as motivation to utilize the RF algorithm in this paper to predict cloud fraction from sky images by classifying cloud and non-cloud pixels.

The process of selecting features from the images under examination is a pivotal step in image processing and in-depth understanding of the scenes observed through the sky imager. These features can be spectral, textural or a combination of both.

The selection done in the paper includes the fundamental red (R), green (G), and blue (B) colour channels, which provide insights into colour composition and distribution within the images. Additionally, the Hue, Value, and Saturation (HVS) model is considered, offering information about the dominant colour tone, brightness, and colour vividness, thus contributing to the interpretation of visual perception. To delve into spectral properties, the ratio of red (R) to blue (B) channels and its logarithmic counterpart are used, revealing variations that are indicative of cloud presence and atmospheric conditions. Notably, the RAS (Removal of Atmospheric Scattering) feature, as introduced by (Yang et al., 2017), emerges as a key component in this segmentation task. This composite parameter mitigates the influence of atmospheric scattering on image data by merging the panchromatic, bright, and dark channels. It minimizes the inhomogeneous sky background throughout the image and thereby enhances the distinction between cloud and sky regions. Each of these features brings a unique perspective to image analysis, encompassing a diverse array of image characteristics, that play distinct and indispensable roles in the decision tree.

To evaluate the computational performance of the proposed model, the RF classifier's inference benchmarks were run on a desktop machine with an Intel Core i7-11700 CPU (8 cores, 16 threads), 16 GB RAM, and no GPU acceleration, running Windows 11 (64-bit). Inference was performed on 280×280-pixel images (~78,400 pixels) with an average runtime of 0.113 seconds per image, a peak memory usage of 41 MB, and an effective processing speed of approximately 800,000 pixels per second. These results reflect the classifier's suitability for real-time, low-power applications without the need for specialized hardware. An overview of the proposed RF based cloud detection pipeline is shown in Fig.1.

Figure 1: Workflow of the Random Forest-based cloud detection framework. Input images are pre-processed and annotated to create a master dataset, which is split into training, validation, and test sets. The Random Forest model is trained with hyperparameter tuning and evaluated on validation data. The trained model generates predicted cloud masks, from which cloud fraction is computed and compared against ground truth for output validation.

## 4 Model Training and Evaluation

For each of the locations using TSI, a set of 300 images were selected of that particular location to train a random forest classifier. While the set of images are a representation of different cloud fractions, they also encompass various cloud types, weather conditions, and lighting scenarios of each location. The classifier was configured with 100 trees and a fixed random seed to ensure the reproducibility of results. A train-test split of 80:20 was applied on the dataset and after training, the model is used to classify cloud and non-cloud pixels of each sky image in the test set.

Each model, trained specifically using images of that location, was used to predict the cloud pixels from the test images corresponding to that location. We computed various performance metrics, including accuracy, F1-score, precision, recall, ROC-AUC score and Intersection over Union (IoU) score to assess the classifier's effectiveness in distinguishing between cloud and non-cloud regions which is tabulated in Table 1. The confusion matrix for each location is provided in Fig.S2 of supplementary along with the description of each performance metric.

Table.1 shows the performance metrics for each dataset location

| Location              | Accuracy | Precision | Recall | F1 Score | ROC-AUC Score | IoU Score |
|-----------------------|----------|-----------|--------|----------|---------------|-----------|
| Black Forest, Germany | 0.93     | 0.88      | 0.88   | 0.88     | 0.88          | 0.79      |
| Lamont, Canada        | 0.89     | 0.84      | 0.90   | 0.87     | 0.85          | 0.76      |
| Darwin, Australia     | 0.91     | 0.88      | 0.93   | 0.90     | 0.87          | 0.80      |
| Gadanki, India        | 0.94     | 0.85      | 0.92   | 0.88     | 0.90          | 0.79      |

While the accuracy score for all locations has been greater than 88%, the F1 score is also hovering around the same figure, suggesting that the model has been trained on a well-balanced dataset with different classes. The RF model is neither overly conservative nor too lenient in predicting cloud pixels as suggested by the precision and recall values lying between 0.84 to 0.88 and 0.88 to 0.93 respectively. The ROC-AUC score is also high across all locations, indicating that the model has good discriminatory ability between different classes (e.g., cloud vs. no cloud). The IoU scores are all above 0.75, indicating a significant overlap between the predicted cloud regions and the ground truth. Overall, the model shows strong predictive ability across different geographical locations.

The model's primary objective is to determine the cloud fraction, representing the proportion of the area covered by clouds relative to the total area. This effectively is a ratio between the number of pixels that are clouds to the total number of visible sky pixels. Thus, the location specific trained model was applied on the corresponding validation set and the predicted cloud and non-cloud pixels were used to measure cloud fraction (CF). The measured CF were compared against the ground truth annotated CF values for the validation set images of each location, providing a direct measure of the model's CF prediction accuracy. A scatter plot of the predicted CF vs the ground truth for each location is shown in Fig. 2.

#### 4.1 Model Validation

Initially, a same-location RF classifier was trained using 300 images from individual sites using the TSI and validated on images from the same site. Subsequently, a unified training set of 300 images was created by combining some of the images from the training sets of all four sites that utilize the TSI. A new, multi-location trained RF model was developed using this merged training set, and its performance was evaluated on the validation set from each site. Additionally, the cloud fraction data provided by TSI of the ARM sites and Gadanki were used for comparison with the RF classifier outputs and the ground truth. The results of the comparative analysis from these experiments are shown in Fig. 2.

Figure 2: Comparison of cloud fraction output against ground truth, of single-location RF model (in blue dots), multi-location RF model (in green squares) and TSI output (in black plus symbol) of ARM sites located at Black Forest Germany, Lamont Canada, and Darwin Australia and Gadanki, India where a TSI has been used to get sky images.

It can be inferred from the graphs that the same-location trained RF classifier has generally outperformed the multi-location trained RF classifier, which is expected. However, the difference in performance is not substantial, suggesting a location

agnostic behaviour of the classifier model. Furthermore, the RF classifier shows better accuracy compared to the TSI output, as indicated by the higher fit value (R<sup>2</sup>) of the RF classifier model in all four cases. This is further illustrated in Fig. 3, which compares the outputs of our RF classifier and the TSI for three randomly selected sky images taken by the TSI at Gadanki, India. In these cases, the TSI struggled to detect all clouds, missing several significant cloud formations accurately. In contrast, our RF classifier performed notably better, closely aligning with the annotated clouds.

220

Figure 3: Comparison of the detected clouds by the RF Classifier and by the TSI (at Gadanki, India) with the annotated clouds. The first column shows the actual images captured by TSI on 3<sup>rd</sup> July 2010 at 10:15 AM, 10:25 AM and 11:00 AM IST. The second column is the corresponding annotated image, with the white colour representing clouds and everything else in black. The third column is the RF classifier's image output of the detected cloud pixels, with white colour representing clouds and everything else in black. The fourth column shows the cloud pixels detected by TSI software, with white colour as thin clouds, grey colour as thick clouds, blue as sky, sun's position as yellow and everything else in black. The TSI is underestimating the cloud pixels in all three cases while the RF classifier is capturing them effectively.

Another intriguing observation that is highlighted in Fig. 2 is the variability of TSI's output over various regions. The TSI outputs exhibit higher accuracy at locations characterized by lower pollution levels, such as Black Forest, Germany, and Lamont, Canada, compared to areas with elevated pollution loading, such as Gadanki, India, and Darwin, Australia. This observation suggests that the TSI may yield more reliable results in cleaner environments due to reduced atmospheric interference and greater clarity of sky images. However, locations with higher pollution levels may introduce complexities and uncertainties in TSI outputs, potentially compromising their accuracy.

230

In contrast, the RF classifier model is relatively unaffected by location-specific pollution loading effects. Regardless of the environmental conditions or pollution levels, the RF classifier maintains its accuracy in estimating cloud fractions from sky images. This robustness highlights the adaptability and generalizability of the RF classifier model across different geographical locations with similar imaging equipment.

The RF classifier is also able to capture the regional trends of cloud fraction across all four locations as evident in Fig. 4. It shows the median CF data as heatmaps with each row corresponding to one of the four locations and each column corresponding to median CF obtained from TSI data, median CF predicted by our RF classifier and the percentage difference between them respectively. The horizontal axis of the heatmap represents the months of the year (January to December) in numbers, and the vertical axis represents the hours of the day from 6 AM to 6 PM local time for each region. The heatmap colour gradient indicates the CF values, with darker shades representing higher cloud fractions. While the general patterns match, subtle regional differences become apparent in the percentage difference heatmaps. In the case of Australia, the TSI overestimates cloud cover in the first half of the year and underestimates it in the latter half. This is seen in the positive percentage differences (warmer colours) in the early months and negative differences (cooler colours) later in the year. Germany and Canada datasets show relatively stable agreement between TSI and RF, with only slight overprediction by TSI. This consistency suggests that the RF model is successfully capturing the general climate and cloud trends for these regions, with TSI performing reasonably well, though slightly skewed toward overprediction. In India, the RF and TSI heatmaps show a stark contrast. The RF classifier predicts higher cloud fractions throughout the year compared to the TSI data.

The percentage difference heatmap for India shows predominantly negative values (cooler colours), indicating that TSI persistently underpredicts the cloud fraction in this region across all months and hours. This consistent underestimation suggests that TSI data struggles to capture Gadanki's cloud dynamics properly throughout the year. In turn, the RF classifier, having been trained on local data, is better adapted to handling the unique cloud patterns seen here.

Figure 4: Median Cloud Fraction (CF) heatmaps for four regions—Australia, Germany, Canada, and India—comparing CF estimates from TSI data, RF classifier output, and their percentage difference. The horizontal axis denotes the months (January to December), and the vertical axis indicates the local time of day (06:00–18:00). Distinct regional patterns emerge: TSI tends to overestimate CF in Australia (January–June) and in Germany and Canada, while underestimating CF in India.

# 4.2 Application of RF Classifier Model over Merak

285

290

A crucial obstacle that has been encountered pertains to the compatibility of the RF model across different imaging equipment. This can be attributed to the inherent variations among sensors used in CMOS cameras and CCD cameras. This leads to discrepancies in the image characteristics such as colour, rendition, contrast, and resolution. Consequently, attempting to generalize the RF model to images from disparate sources becomes impractical due to the divergence in sensor specifications and calibration methodologies. That is why the RF classifier developed using the TSI image data cannot be used for Merak, which uses a CMOS-sensor based all-sky imager.

Thus, the images of Merak underwent a similar process of data selection, training, testing, and validation to create a different model, specific to this location. After getting a good accuracy score of 95% for the test dataset, the model's effectiveness was verified by employing it to predict the cloud fraction for the validation set images. The scatter plot of the predicted cloud fraction vs the ground truth in Fig. 5(a) shows a good fit of about 0.98 with a root mean squared error as low as 0.05. This substantiates the high accuracy score of 95% and serves as verification of the model's effectiveness at a different location with a different imaging instrument. A few accurately predicted outputs of the RF classifier have been shown in Fig. S3 of the supplementary document.

Despite a good fit between the ground truth and the predicted output, as evident from Figs. 2 and 5(a), there are a few points in the plot that have significant disparities. A few of these disparities are illustrated in Fig. 5(b). A significant source of error affecting the predicted output can be attributed to sun glare and cirrus clouds. Although naturally occurring and often unavoidable, these elements introduce complexities and uncertainties that can pose challenges for accurate image analysis and interpretation. Sun glare often leads to overexposed or saturated pixels, making it challenging to extract meaningful information about the sky's properties. As a result, an inaccuracy is introduced in cloud detection.

Cirrus clouds, on the other hand, add a layer of complexity due to their intricate filamentous structure and high altitude. These clouds, composed of ice crystals, present unique challenges for accurate classification and quantification. Their thin and translucent nature can make them challenging to distinguish from the background, especially when they partially obscure other cloud types or the sun. Consequently, annotation errors arise in interpreting cirrus clouds and achieving precise semantic annotations becomes a labour-intensive task. Therefore, the presence of cirrus clouds can lead to both false positives and false negatives in cloud detection, impacting the overall quality of cloud fraction estimates. Nonetheless, it's worth noting that in the validation set of 500 images of Merak India, 1.6% of the images had cirrus clouds, with a mean CF error of  $0.14 \pm 0.04$ . Similarly, about 4.2% of the validation set had sun glare with mean CF error of  $0.12 \pm 0.02$  as evident from Fig.5(c). These errors collectively account for a very low percentage of the overall dataset, making them relatively insignificant in the broader context.

Figure 5: (a) Validation of RF classifier output for images taken at Merak, India (b) Representative failure cases: top row shows overprediction due to sun glare (highlighted by red circle in (a)), and the bottom row shows underprediction caused by cirrus clouds (highlighted by blue square in (a)). Red and blue pixels in the difference column, indicate misclassified pixels. (c) violin plot that compares CF errors for cirrus and sun glare cases

## Conclusion

CF is an essential climate variable required by the scientific community for studying climate change. It has numerous practical applications, including studying the Earth's radiation budget, predicting future climate patterns, monitoring agricultural activities, forecasting solar energy, and assessing resources. Additionally, cloud cover data is used as input in models for studying pollution and climate. While traditional cloud segmentation techniques often rely on empirically determined threshold values, their accuracy falters under complex atmospheric and cloud conditions. This study explores the efficiency of the Random Forest (RF) classifier in pixel-wise cloud segmentation, using a well-curated dataset of semantically annotated images from five different locations. Training data with diverse sky conditions and atmospheric loading, collected over a year for each location, was meticulously selected. Subsets of these training images were used for rigorous model evaluation across multiple metrics.

The RF classifier demonstrates strong predictive ability across all locations, with accuracy and F1 scores consistently above 88%, indicating a well-balanced dataset. High ROC-AUC scores of more than 0.85 and IoU scores of more than 0.79 further confirm the model's robust discriminatory ability between cloud and non-cloud classes. Additionally, the RF classifier showed strong accuracy and fit metrics, particularly in locations with high pollution levels, such as India and Australia. The model's ability to generalize across diverse geographic sites highlights its location-agnostic nature, maintaining high performance even when trained on mixed datasets from multiple regions. Furthermore, the RF classifier demonstrated superior capability in capturing long-term weather and cloud patterns, making it a valuable tool for estimating cloud cover and broader climate studies.

However, the model did encounter challenges in handling sun glares caused by incomplete shadow band coverage and distinguishing cirrus clouds, primarily due to annotation limitations. These shortcomings, while noteworthy, represent a minor fraction of the overall dataset (roughly 6%), and their impact on CF estimates remain minimal. Overall, the RF classifier proves to be a highly effective and adaptable tool for cloud segmentation, with significant potential for improving cloud cover analysis, especially in regions with complex atmospheric conditions.

# Data and Code Availability:

All TSI sky images are publicly available at <a href="https://adc.arm.gov/discovery/#/results/instrument\_class\_code::tsi">https://adc.arm.gov/discovery/#/results/instrument\_class\_code::tsi</a> and their cloud fractions are available at <a href="https://adc.arm.gov/discovery/#/results/primary\_meas\_type\_code::cldfraction">https://adc.arm.gov/discovery/#/results/primary\_meas\_type\_code::cldfraction</a>. The sky images of Merak and Gadanki, India, along with all annotations and code, can be provided on request to the corresponding author.

#### **Author contribution:**

Conceptualization, Methodology and Formal analysis: CS, NP and SKS.

325 Data curation and software: SKS, BLM, SSN, RB, MVR

Supervision: CS, NP

Writing - original draft preparation: SKS, CS and NP

Writing – review & DLM, SSN, RB, MVR

# **Acknowledgement:**

We thank Tundup Stanzin, Tsewang Punchok, Tundup Thinless, Stanzin Norbo, Kunzhang Paljor, Konchok Nimaand, and Namgyal Dorje of the National Large Solar Telescope project at Merak station for their help in downloading the data from the all-sky camera instrument and maintaining the database. We also thank Volam Santhosh Kumar and Aninda Bhattacharya for helping in annotation of the images to generate the ground truth.

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
