# Peer review of "Cloud Fraction estimation using Random Forest classifier on Sky Images"

_EGUsphere, 2024_

## Author Comment (AC1)

**General comments**

================

**The paper presents cloud fraction estimates using a random forest classifier on sky images. They compare the prediction results of the model against the semantically annotated sky images originating from multiple observation sites in the world. Overall, the paper is well structured and well explained; making it easier for the reader to follow and understand. The authors emphasize on the limitations of the study that are (1) the white-balance / colour calibration of the datasets acquired with different sensors; (2) the air pollution levels that prevents more accurate predictions for India and Australia data; (3) the artifacts on the images (sun glares, cirrus) making it difficult to extract information on sky properties. Despite these, they demonstrated that RF classifier outperforms traditional methods in high-pollution sites. Minor corrections need to be applied to improve readability, structure and syntax.**

We sincerely appreciate the reviewer's assessment and encouragement towards the structure and clarity of our study. We also understand the concerns raised about the limitations of the model. Accordingly, we have addressed all the concerns and our response to the specific comments, documented below. The **bold text in black represents the reviewer's comments**, followed by our responses in blue colour regular text, and the *italic texts in red indicate the corresponding changes made in the main manuscript*.

Specific comments

==================

**Introduction:**

**The introduction lacks a presentation/structure paragraph of the remaining of the sections of the paper. Like : "The paper is structured as follows. In Section 2 we present..."**

**Response:**

We thank the reviewer for bringing this to our attention. The manuscript has been updated with the following additional lines in the introduction section:

*To support these objectives, the remainder of the paper is structured as follows. Section 2 describes the data used in this study, including observing sites, datasets, preprocessing steps, and the creation of ground truth masks. Section 3 details the model selection process. Section 4 covers model training and evaluation, with Section 4.1 focusing on model validation and Section 4.2 discussing the application of the Random Forest classifier over the Merak site. Finally, Section 5 presents the conclusions drawn from this study.*
* * *
**Section 2**

**Maybe merge the Section 2 "Observing sites" and 3 "Data Generation and Preprocessing" into one single section called "Data" for easier structure for the reader:**

**2. Data**

   **2.1 Observing sites and datasets**

**2.2 Preprocessing**

**2.3 Selection**

**2.4 Ground truth masks**

**Response:**

We thank the reviewer for the suggestion. We have made changes in the updated manuscript accordingly.
* * *
**Section 3**

**Add a 2 panels figure to show a raw image vs the preprocessed version, to illustrate where the dead zones are located in the image, and what the circular mask does.**

**Response:**

We thank the reviewer for the suggestion. However, instead of adding the figure to the main manuscript, we have included it in the supplementary figures for better readability. The figure has been attached below for your reference.

[Figure]

*Fig.S1. Left side - Raw image (from Lamont Canada) captured at 5:07 PM on 07-04-2010 showing image of sky along with the surrounding area that is not essential for cloud fraction estimation. Right side – Processed image after removing dead zones by cropping and resizing the raw image and then applying a circular mask to highlight only the usable area of the raw image.*
* * *
Section 4

**Figure 1 must be cited explicitly in the text and lacks a caption sentence.**

**Response:**

We thank the reviewer for bringing this to our attention. In the revised version, we have provided a more detailed and informative caption as follows:

[Figure]

*Fig.1. Workflow of the Random Forest-based cloud detection framework. The input images are pre-processed and annotated to create a master dataset, which is then split into training, validation, and test sets. The Random Forest model is trained with hyperparameter tuning and evaluated on validation data. The trained model generates predicted cloud masks, from which cloud fraction is computed and compared against ground truth for output validation.*

We have also cited the figure in the text in section 3 as follows:

*An overview of the proposed RF based cloud detection pipeline is shown in Fig.1.*
* * *
Section 5

**Any idea/hypothesis of why the model performs worse in Canada and Australia ? You stated some explanations for Indian data, but not for these. Why is has the German dataset the best results ?**

**Response:**

We appreciate the reviewer's concern. However, based on our analysis, we do not observe a significantly poorer performance over Canada or Australia compared to Germany. The relatively better performance of both the same-location and multi-location RF models, as well as the TSI, over Germany can be attributed to the site's comparatively cleaner atmospheric conditions. This is evident from Supplementary Fig. S4 (attached below for reference), which shows that the mean aerosol

optical depth (AOD) over Germany is consistently lower than at the other sites. The lower AOD likely reduces the complexity of aerosol-radiation interactions, leading to improved model performance in this region.

[Figure]

*Fig.S4. Box plot of aerosol optical depth (AOD) at 500nm at the three ARM sites, i.e. Black Forest, Germany; Lamont, Canada; and Darwin, Australia. The plot shows the occurrence of higher AOD in Canada and Australia compared to Germany.*
* * *
Section 5.1

The first paragraph needs to be moved in a prior position in the paper as it explains what is the cloud fraction and what's the goal of the model of this study; like at the beginning of Section 4 for example, or even earlier in the paper.

**Response:**

We thank the reviewer for the suggestion. We have moved the paragraph to section 4 in the updated manuscript with the following changes:

*The model's primary objective is to determine the cloud fraction, representing the proportion of the area covered by clouds relative to the total area. This effectively is a ratio between the number of pixels that are clouds to the total number of visible sky pixels. Thus, the location specific trained model was applied on the corresponding validation set, and the predicted cloud and non-cloud pixels were used to measure cloud fraction (CF). The measured CF was compared with the ground truth annotated CF values for the validation set images of each location, providing a direct measure of the model's CF prediction accuracy. A scatter plot of the predicted CF vs the ground truth for each location is shown in Fig. 2.*
* * *
Section 5.2

- First paragraph : **accurate white-balance or colour calibration is required then ? Is there some practical solution to this sensor uniformization issue ?**

**Response:**

We thank the reviewer for bringing this point to our attention. However, achieving uniform colour calibration across all types of imaging sensors is extremely challenging due to differences in sensor characteristics, optics, and internal design of the manufacturer. While normalization techniques can help reduce variability, complete uniformization is not always feasible—particularly when dealing with two different types of imaging sensors (namely CCD and CMOS that work on completely different mechanisms).
* * *
**- Add a Figure or subfigure for which the predictions are really accurate, instead of only showing the shortcomings and outliers.**

**Response:**

We thank the reviewer for the suggestion. We have added some of the accurately predicted images in the supplementary Fig. S3, as well as attached below.

[Figure]

*Fig.S3. A few predicted accurate outputs of the RF classifier for images, at Merak, India,*
* * *
**Technical corrections**

=====================

**Abstract:**

**- "vary" used too much in 3 consecutive sentences; use synonym**

**Response:**

The manuscript has been updated with the correction.

**- efficiency instead of efficacy**

**Response:**

The manuscript has been updated with the correction.

**Introduction:**

**- line 27-28: instead of we need, use "the scientific community requires specific devices"**

**Response:**

The manuscript has been updated with the suggestion.

**- line 31 : capture data at high temporal resolutions**

**Response:**

The manuscript has been updated with the correction.

**- line 36 : caveat with " Many researchers have adopted this (Chauvin et al., 2015; Chow et al., 2011; Ghonima et al., 2012; Kuhn et al., 2018);(Lothon et al., 2019)" and then comma and end sentence. Something is lacking. Replace by:**

**The clear sky (CSL) threshold method, as outlined by Shields et al. (2009), uses spectral information—particularly from the red and blue bands—to differentiate between cloudy and clear-sky conditions. This technique has been widely adopted by researchers (e.g., Chauvin et al., 2015; Chow et al., 2011; Ghonima et al., 2012; Kuhn et al., 2018; Lothon et al., 2019). However, a notable limitation is that the threshold value can vary across an image, influenced by the relative distance between the sun and each image pixel.**

**Response:**

The manuscript has been updated with the suggested sentence.

**- line 40 : use citet or citealt and not citep for in-sentence citation.**

**Response:**

The manuscript has been updated with the correction.

**- line 46 and so on : same citation issue**

**Response:**

The manuscript has been updated with the correction.

**- line 57 : distinguish clouds from clear sky**

**Response:**

The manuscript has been updated with the correction.

Section 2

**- line 78 : reformulate better "All these sites have the common make of instrument - a Total Sky Imager (TSI) that takes the sky images."**

**Response:**

We thank the reviewer for the suggestion. Here is the updated sentence in the manuscript:

*All four sites utilize the same type of instrument - a Total Sky Imager (TSI) - for capturing sky images.*

**- line 81 : syntax + repetition "A shadow band is also placed that continuously rotates as it tracks the sun. This shadow band blocks the intense direct sun that can saturate the images."**

**Response:**

Here is the updated sentence in the manuscript:

*To prevent image saturation from direct sunlight, a rotating shadow band is used to continuously track and block the sun.*

Section 3.1

**- line 108 : "this would be essential" and not instrumental**

**Response:**

The manuscript has been updated with the correction.

**- line 110 : validation of the models training.**

**Response:**

The manuscript has been updated with the correction.

Section 3.2

**- MATLAB image labeller app: add some footnote link or citation**

**Response:**

The suggestion has been added to the updated manuscript.

Section 4

**- line 130 + 131 : replace no. with the number of**

**Response:**

The manuscript has been updated with the correction.

**- line 131 : did not explain what Mfeat is**

**Response:**

We have defined $M_{feat}$ as the number of selected feature variables in the updated manuscript.

**- line 137 : reformulate like for example:**

**"The primary limitation of Random Forest (RF) models lies in their interpretability. Because RF relies on an ensemble of numerous decision trees to generate predictions, it can be challenging to trace and explain the rationale behind a specific prediction."**

**Response:**

We thank the reviewer for the suggestion. However, to avoid any confusion and increase readability, we have changed the sentence as follows:

*A key limitation of Random Forests is that, due to their ensemble nature, it is difficult to trace individual pixel-level classifications back to specific features or decision paths.*

**Section 5**

**- line 161 : no comma before "were used to train a random forest classifier"**

**Response:**

We thank the reviewer for bringing this to our attention. The manuscript has been updated with the correction.

**- line 162 : no need for (n_estimators=100) and (random_state=42)**

**Response:**

We thank the reviewer for the suggestion. The manuscript has been updated accordingly.

**- line 165 : let's call it the "test" or "validation" dataset ?**

**Response:**

We thank the reviewer for the suggestion. The manuscript has been updated accordingly.

**- line 170 : "Table 1 shows the metrics for each dataset location"**

**Response:**

We thank the reviewer for the suggestion. The manuscript has been updated accordingly.

- line 175 : "Overall, ..."

**Response:**

The manuscript has been updated with the correction.

Section 5.1

**- Fig 2 : last two sentences need to be moved in the main text after the reference of the Figure or discarded to avoid repetitions in the paper.**

**Response:**

We thank the reviewer for the suggestion. The manuscript has been updated accordingly, with the last two sentences discarded to avoid repetition.

**- Fig. 2 : To improve clarity, use different symbols for each model, e.g. dots, crosses and triangles. FYI the plots of Fig. 2, do not appear in color on the prepublished version.**

**Response:**

According to your suggestion, we have made changes in the figure and updated the manuscript as follows:

[Figure]

*Figure 2: Comparison of cloud fraction output against ground truth, of single-location RF model (in blue dots), multi-location RF model (in green squares) and TSI output (in black plus symbol) of ARM sites located at Black Forest (Germany), Lamont (Canada), and Darwin (Australia), and Gadanki (India), where a TSI has been used to get sky images.*

**- Fig 3 : line 205 : "first column are" English**

**Response:**

The manuscript has been updated with the correction.

**- Fig 3 : "nicely" is not to be used in the text; use more objective synonym**

**Response:**

The manuscript has been updated with the word *"effectively"*.

**- line 215 : do you have citation that maybe studies this phenomena ?**

**Response:**

Thank you for your observation. Currently, we are not aware of any prior studies that directly investigate this phenomenon. However, in our analysis, we observed that sites with consistently higher aerosol optical depth (AOD) values—such as Lamont (Canada), Darwin (Australia), and Gadanki (India)—tend to show less accurate Total Sky Imager (TSI) output compared to a lower AOD site like Germany. This observation is illustrated in the figure below, which presents a box plot of AOD at 500 nm for the three ARM locations.

While this pattern suggests a potential relationship between elevated aerosol loading and reduced TSI performance, a more rigorous and dedicated study is needed to confirm and explain this interaction. We consider this an important direction for future work.

[Figure]

*Fig.S4. Box plot of aerosol optical depth (AOD) at 500nm at the three ARM sites, i.e. Black Forest, Germany; Lamont, Canada; and Darwin, Australia. The plot shows the occurrence of higher AOD in Canada and Australia compared to Germany.*

**- line 224 : the horizontal axis, instead of "x axis"; idem for vertical axis; January to December in full words**

**Response:**

The manuscript has been updated with the correction.

**- line 229 : "Germany and Canada" datasets, to precise**

**Response:**

The manuscript has been updated with the correction.

**- Fig 4 : reformulate better like:**

**"Figure 4: Median Cloud Fraction (CF) heatmaps for four regions—Australia, Germany,**

Canada, and India—comparing CF estimates from TSI data, RF classifier output, and their percentage difference. The horizontal axis denotes the months (January to December), and the vertical axis indicates the local time of day (06:00–18:00). Distinct regional patterns emerge: TSI tends to overestimate CF in Australia (January–June) and in Germany and Canada, while underestimating CF in India."

**Response:**

We thank the reviewer for the suggestion. We have updated the manuscript as per the suggestion.

**Conclusion**
**- line 274 : "it has numerous..." and not the data as we are talking about the CF**

**Response:**

The manuscript has been updated with the correction.

---

## Author Comment (AC2)

**This paper applies a RF classifier to ground-based sky images from several locations for estimating cloud fraction. The authors prepare annotated datasets, train site-specific and merged RF models, and compare the model's CF output against TSI results. While the manuscript is well-organised and the topic is of interest for the atmospheric observation and machine learning communities, I find the work lack of methodological novelty, analysis depth, and evaluation rigor. The comments below aim to guide the authors toward a significantly strengthened version of this manuscript.**

We sincerely appreciate the reviewer's critical assessment and encouragement towards the technical rigor of our study. We also understand the concerns raised about the evaluation of the model and better quantification of the biases. Accordingly, we have addressed all the concerns and our response to the specific comments, documented below. The **bold text in black represents the reviewer's comments**, followed by our responses in blue colour regular text, and the *italic texts in red indicate the corresponding changes made in the main manuscript.*
* * *
**1. The model is explicitly trained at the pixel level (as shown in Fig. 3), yet the evaluation is based solely on cloud fraction (CF), a scene-level aggregate statistic. This disconnect is concerning. If the model is trained to perform per-pixel segmentation, why are there no pixel-wise metrics (e.g., accuracy, F1, precision/recall, IoU) reported? This omission makes it difficult to assess how well the model actually distinguishes cloud vs. sky on a per-pixel basis, and not just whether it approximates CF correctly. Especially given that annotated segmentation masks are available, this should be straightforward to add.**

**Response:**

Yes, the model is trained at the pixel level and our evaluation also include pixel-wise metrics. As per the reviewer's concern, we have also revised the relevant sentences in the manuscript (Line nos 161 to 170) and Table 1 to explicitly state about this clarification.

The values reported in Table 1 - namely, accuracy, F1-score, and ROC-AUC, are all computed at the pixel level using the predicted cloud/non-cloud masks against the ground truth. These metrics directly reflect the per-pixel classification performance, consistent with the model's pixel-wise training shown in Fig. 1. The cloud fraction (CF) metric was also reported to provide a scene-level perspective on the model's behaviour, as CF is often used in climate-related or radiative transfer applications.

We have now added the precision, recall and IoU scores of all the locations in Table 1 in the revised manuscript and provided the confusion matrix for the same in the supplementary as Fig.S2. The revised text is mentioned below for the reviewer's reference.

*For each of the locations using TSI, a set of 300 images was selected for that particular location to train a random forest classifier. While the set of images is a representation of different cloud fractions, they also encompass various cloud types, weather conditions, and lighting scenarios of each location. The classifier was configured with 100 trees and a fixed random seed to ensure the reproducibility of results. A train-test split of 80:20 was applied to the dataset, and after training, the model was used to classify cloud and non-cloud pixels of each sky image in the test set.*

*Each model, trained specifically using images of that location, was used to predict the cloud pixels from the test images corresponding to that location. We computed various performance metrics, including accuracy, F1-score, precision, recall, ROC-AUC score and Intersection over Union (IoU) score to assess the classifier's effectiveness in distinguishing between cloud and non-cloud regions, which is*

*tabulated in Table 1. The confusion matrix for each location is provided in Fig.S2 of the supplementary, along with the description of each performance metric.*

*Table.1 shows the performance metrics for each dataset location*

| *Location* | *Accuracy* | *Precision* | *Recall* | *F1 Score* | *ROC-AUC Score* | *IoU Score* |
|---|---|---|---|---|---|---|
| *Black Forest, Germany* | *0.93* | *0.88* | *0.88* | *0.88* | *0.88* | *0.79* |
| *Lamont, Canada* | *0.89* | *0.84* | *0.90* | *0.87* | *0.85* | *0.76* |
| *Darwin, Australia* | *0.91* | *0.88* | *0.93* | *0.90* | *0.87* | *0.80* |
| *Gadanki, India* | *0.94* | *0.85* | *0.92* | *0.88* | *0.90* | *0.79* |

***In Supplementary:***

[Figure]

*Fig.S2. Confusion matrix obtained from the test set for each location.*
* * *
**2. Though the authors mention performance degradation due to sun glare and cirrus clouds, this is illustrated only through a few hand-picked examples. There is no quantification of how prevalent these issues are in the dataset, nor an analysis of how performance varies across such conditions. Similarly, no confusion matrix or class-specific breakdown is presented to identify key failure modes. A more systematic error analysis would strengthen this part.**

**Response:**

We thank the reviewer for this thoughtful suggestion. We agree that a systematic error analysis would strengthen the discussion of model limitations. In our current manuscript, we included representative failure cases (e.g., due to sun glare and cirrus clouds) to illustrate challenging scenarios qualitatively. Moreover, as per your suggestion, we have added additional analysis on the CF errors caused if cirrus clouds or sun glare (figure attached below for reference).

Among the 500 images in the validation set of Merak, 1.6% of the images had cirrus clouds with a mean CF error of $0.14 \pm 0.04$. About 4.2% of the validation set had sun glare with a mean CF error of $0.12 \pm 0.02$. We have also updated our manuscript to highlight these errors.

[Figure]

Figure 5: (a) Validation of RF classifier output for images taken at Merak, India (b) Representative failure cases: top row shows overprediction due to sun glare (highlighted by red circle in (a)), and the bottom row shows underprediction caused by cirrus clouds (highlighted by blue square in (a)). Red and

*blue pixels in the difference column, indicate misclassified pixels. (c) violin plot that compares CF errors for cirrus and sun glare cases*
* * *
**3. The authors highlight that RF is computationally efficient, but there is no measurement of runtime, memory usage, or inference speed. Even a simple runtime comparison on a CPU vs. a lightweight CNN would be informative.**

**Response:**

We thank the reviewer for this helpful observation. The statement that the Random Forest (RF) model is computationally efficient was to highlight a known advantage of RFs as established in prior literature (Mu et al., 2017). Our primary focus was on demonstrating that an RF, when trained with appropriate features, can achieve competitive performance in cloud detection tasks.

However, as per reviewer's suggestion, we performed some inference benchmarks for the RF classifier. We have revised the manuscript with the following information:

*To evaluate the computational performance of the proposed model, the RF classifier's inference benchmarks were run on a desktop machine with an Intel Core i7-11700 CPU (8 cores, 16 threads), 16 GB RAM, and no GPU acceleration, running Windows 11 (64-bit). Inference was performed on 280×280-pixel images (~78,400 pixels) with an average runtime of 0.113 seconds per image, a peak memory usage of 41 MB, and an effective processing speed of approximately 800,000 pixels per second. These results reflect the classifier's suitability for real-time, low-power applications without the need for specialized hardware.*

….

Reference:

Mu, X., Ting, K. M., and Zhou, Z.-H.: Classification Under Streaming Emerging New Classes: A Solution Using Completely-Random Trees, IEEE Trans Knowl Data Eng, 29, 1605–1618, https://doi.org/10.1109/TKDE.2017.2691702, 2017
* * *
**Other minor comments:**

**Line 30: "satellite-based imagers have lower temporal resolutions". The authors ignore the fact that geostationary satellites provide very high temporal resolution (10-minute or better) imagery. This should be acknowledged to give a more balanced view.**

**Response:**

We thank the reviewer for bringing this to our attention. We have revised the statement as follows:

*Satellite imagers observe clouds over larger spatial domains (Verma et al., 2018), often with temporal resolutions as good as 10 mins (Huang et al., 2019).*

References:

Huang, Y. I., Siems, S., Manton, M., Protat, A., Majewski, L., and Nguyen, H.: Evaluating himawari-8 cloud products using shipborne and CALIPSO observations: Cloud-top height and cloud-top temperature, J Atmos Ocean Technol, 36, 2327–2347, https://doi.org/10.1175/JTECH-D-18-0231.1, 2019.

Verma, S., Rao, P. V. N., Shaeb, H. B. K., Seshasai, M. V. R., and PadmaKumari, B.: Cloud fraction retrieval using data from Indian geostationary satellites and validation, Int J Remote Sens, 39, 7965–7977, https://doi.org/10.1080/01431161.2018.1479792, 2018.
* * *
**Line 95: "images captured during rain were also removed". Please clarify how rain-contaminated images were identified. Was this done manually or through an automated threshold/filter?**

**Response:**

We thank the reviewer for this important clarification request. The identification of rain-contaminated images was performed manually, based on visible artifacts such as raindrops on the lens, severe blurring, or overall low visibility that typically accompany rain events. These images were visually inspected and excluded during dataset curation to ensure the model was trained only on usable sky conditions.

We have revised the manuscript to explicitly state that the removal process was done through manual inspection.
* * *
**Line 123: "Random Forest" -> Should be abbreviated as RF.**

**Response:**

The manuscript has been updated with the correction.
* * *
**Line 137: The sentence stating that RF models are "difficult to interpret" is vague. Please be specific: are the authors referring to the difficulty of tracing individual pixel classifications back to specific trees or features? If so, mention this explicitly.**

**Response:**

We thank the reviewer for this insightful observation. We have revised the sentence as follows for more clarity:

*A key limitation of Random Forests is that, due to their ensemble nature, it is difficult to trace individual pixel-level classifications back to specific features or decision paths.*
* * *
**Line 159 (Figure 1 caption): The caption is too terse. I would expect a more informative caption that explains the key steps in the algorithm flowchart.**

**Response:**

We thank the reviewer for bringing this to our attention. We have revised the caption as below, to provide a more detailed and informative caption that clearly explains the data flow, training pipeline, and evaluation steps of the proposed methodology as shown in the flowchart.

Figure Caption:

*Fig.1. Workflow of the Random Forest-based cloud detection framework. The input images are pre-processed and annotated to create a master dataset, which is then split into training, validation, and test sets. The Random Forest model is trained with hyperparameter tuning and evaluated on validation data. The trained model generates predicted cloud masks, from which cloud fraction is computed and compared against ground truth for output validation.*